# Investigation of Primary Carbides in a Commercial-Sized Electroslag Remelting Ingot of H13 Steel

**Xijie Wang** [1,2]**, Guangqiang Li** [1,2] 🆔**, Yu Liu** [1,2]**, Yulong Cao** [1,2]**, Fang Wang** [3] **and Qiang Wang** [1,2],*🆔

[1] The State Key Laboratory of Refractories and Metallurgy, Wuhan University of Science and Technology, Wuhan 430081, China; wangxijie@wust.edu.cn (X.W.); liguangqiang@wust.edu.cn (G.L.); liuyu629@wust.edu.cn (Y.L.); caoyulong@wust.edu.cn (Y.C.)

[2] Key Laboratory for Ferrous Metallurgy and Resources Utilization of Ministry of Education, Wuhan University of Science and Technology, Wuhan 430081, China

[3] School of Metallurgy, Northeastern University, Shenyang 110819, China; wangfang@smm.neu.edu.cn

\* Correspondence: wangqiangwust@wust.edu.cn

**Abstract:** The characteristics of primary carbides in a commercial-sized (one ton) electroslag remelting (ESR) ingot of AISI H13 steel were investigated. The interaction between the primary carbides and inclusions was also clarified. The results indicate that there are two types of primary carbides, V-rich and Mo-rich primary carbides, in the H13 ESR ingot. The quantity, the area fraction, and the size of the two primary carbides tend to decrease from the center of the H13 ESR ingot to the outer surface. Additionally, the V-rich primary carbide is obviously larger than the Mo-rich primary carbide. The $Al_2O_3$ inclusion can promote the precipitation of the V-rich primary carbide, while the MnS inclusion encourages the precipitation of Mo-rich primary carbide. The $CaO \cdot Al_2O_3$ inclusion cannot act as the nucleation site for the precipitation of the two primary carbides. The solid fraction that the V-rich primary carbide begins to precipitate ranges from 0.965 to 0.983, and that for the Mo-rich primary carbide and the MnS inclusion change from 0.9990 to 0.9998 and from 0.989 to 0.990, respectively.

**Keywords:** H13 steel; electroslag remelting; primary carbide; inclusion; solidification

## 1. Introduction

AISI H13 steel is widely used as the raw material of die casting, forging, punching, and extruding tools for its high hardness, high toughness, and good thermal fatigue resistance [1,2]. Since these tools are always employed in harsh working conditions, such as high temperature and heavy impact, the high-quality raw material, i.e., AISI H13 steel, is of vital importance to extend the service life of these tools [3,4].

Electroslag remelting (ESR), as a kind of secondary refining technology, is commonly used for manufacturing alloy steels such as AISI H13 steel for its strong ability to improve the cleanliness and enhance the solidification structure [5–7]. Figure 1 shows a schematic of the ESR process. A consumable electrode, the composition of which is the same as the AISI H13 steel, is first prepared by mold casting and hung on the ram. An alternating or direct current is applied to the consumable electrode, which then travels through the molten slag, metal pool, and solidified ingot in a water-cooled copper mold with an open-air or inert atmosphere, and finally arrives at the water-cooled baseplate. The electric current creates enough Joule heating in highly resistive molten slag for melting the consumable electrode. Dense metal droplets sink through the less dense molten slag afterwards, creating a liquid metal pool in the mold. Solidification simultaneously occurs within the liquid metal pool due to the heat is

continuously extracted by the cooling water. A sound AISI H13 steel ingot is therefore formed after the refining process [8].

Due to the microsegregation, carbon would be enriched within the interdendritic region and tend to combine with strong carbide-forming alloying elements during solidification, resulting in the precipitation of primary carbides in the H13 ESR ingot [9,10]. The primary carbide, which is hardly eliminated by regular heat treatment, is considered to have a negative effect on the properties of H13 steel. Not only could the primary carbide decrease the strength, the toughness, and the fatigue performance, but also the primary carbide would act as the initiation site of breakage during the machining process [11–13].

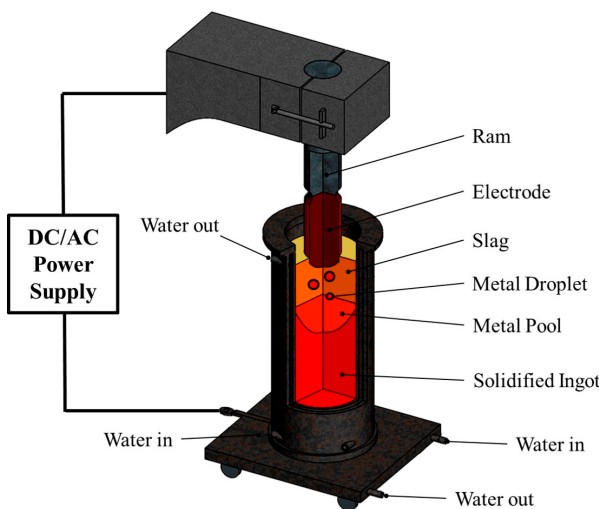

**Figure 1.** Schematic of ESR process (online version in color).

Mao et al. studied the influence of the cooling rate on the primary carbide and found that a higher cooling rate would contribute to the decrease of size and area of primary carbide [14]. The addition of alloy elements, such as magnesium and cerium, were also confirmed to be beneficial for the refining of the primary carbide in the AISI H13 steel [10,15]. It was also reported that the $Al_2O_3$ and $MgO \cdot Al_2O_3$ inclusions could provide the nucleation site for the primary carbide and promote the precipitation of the primary carbide. After calcium treatment, these oxide inclusions with a high melting point would be modified to liquid $CaO \cdot Al_2O_3$ inclusions or partially liquid $CaO \cdot Al_2O_3 \cdot MgO$ inclusions. As a consequence, these modified inclusions would no longer act as the nucleation site for the primary carbide [16].

Though a great deal of effort has been made to obtain a better understanding of the primary carbide in H13 steel, the study of the primary carbide in a commercial-sized AISI H13 ESR ingot is quite rare. Additionally, the relationship of the primary carbide and inclusions in the H13 steel lack of a comprehensive analysis. As stated above, some kinds of inclusion can provide the nucleation site to the primary carbide and promote its nucleation. Thus, it is meaningful to determine the relationship between the primary carbide and inclusions. Due to this, the authors were motivated to investigate the characteristics of the primary carbides and their relationship with the inclusions in a commercial-sized (one ton) AISI H13 ESR ingot.

## 2. Experimental Procedure

### 2.1. Preparation of the ESR Ingot

Reclaimed steel scrap was adopted to manufacture the new AISI H13 ESR ingot. A 15 MW three-phase electric arc furnace was first employed to melt 5500 kg of steel scrap. Three kilograms of aluminum particles were added for deoxidation before the molten steel was poured into a ladle.

The ladle refining was then carried out with $CaF_2$- and CaO-based slag. Meanwhile, the concentrations of alloying elements were adjusted during the refining process. In order to eliminate the dissolved gases in the molten steel, especially hydrogen and nitrogen, a vacuum degassing system was employed. The refined molten steel was cast into five ingots, and each ingot was around 1100 kg with a 200 mm diameter.

An as-cast ingot was then arbitrarily chosen as the consumable electrode, and was loaded upside down in the ESR device after surface grinding. The ESR process was conducted using a water-cooled copper mold with an open-air atmosphere. The inner diameter, height, and wall thickness of the mold are 350 mm, 600 mm, and 65 mm, respectively. The applied current was 5000 A with a frequency of 50 Hz. Additionally, the composition of the slag used in the ESR process was composed of 70 wt% $CaF_2$ and 30 wt% $Al_2O_3$. The weight of the slag was 38 kg. The slag powder was heated for 4 h at 1123 K in a muffle furnace to ensure the removal of moisture from the slag. The AISI H13 ESR ingot, with a 350 mm diameter, 1300 mm length, and weight of one ton, was cooled in a sand pile for 72 h. The chemical compositions of the consumable electrode and the ESR ingot are listed in Table 1 (T. O stands for the total oxygen in steel).

**Table 1.** Chemical compositions of the consumable electrode and the ESR ingot, mass%.

| Ingot | C | Si | Mn | V | Mo | Cr | Al | T. O | S |
|---|---|---|---|---|---|---|---|---|---|
| Electrode | 0.41 | 0.97 | 0.36 | 1.05 | 1.32 | 5.27 | 0.016 | 0.0046 | 0.0080 |
| ESR Ingot | 0.41 | 0.86 | 0.36 | 1.05 | 1.32 | 5.27 | 0.018 | 0.0038 | 0.0040 |

## 2.2. Specimen Preparation

Two plates were taken from the ESR ingot as shown in Figure 2. Both the distance of the upper plate from the top and the distance of the lower plate from the bottom are equal to the 10% of the height of the ESR ingot. Three samples were collected from the center, the mid-radius, and the outer surface of each plate, respectively. The six samples were numbered from S1 to S3 (upper plate), and from S4 to S6 (lower plate). The samples were then carefully polished and etched with 4% nital.

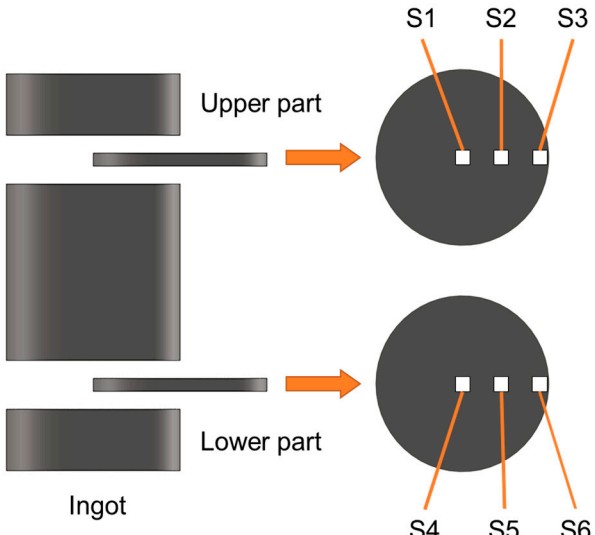

**Figure 2.** Schematic of sampling procedure (online version in color).

## 2.3. Analyzing Approaches

The morphology and composition of the primary carbides and inclusions in the steel matrix were analyzed by scanning electron microscope (SEM, Nova 400 Nano, FEI Co., Oregon, United States of America) equipped with energy dispersive X-ray spectroscopy (EDS, Le350 PentaFETx-3,

Oxford Instruments Co., Oxford, United Kingdom). The primary carbides were investigated from 50 randomly-selected visual fields of every sample. The number, the area, and the size of the primary carbides were counted by Image-Pro Plus 6.0 (Media Cybernetics, Inc., Maryland, United States of America) commercial software. Additionally, the microstructure of the samples was observed using optical microscopy (OM, DSX 510, Olympus Co., Tokyo, Japan).

## 3. Results and Discussion

### 3.1. Characteristics of Primary Carbide

The primary carbides in the H13 ESR ingot were first recognized with the SEM back-scatter mode examination. As shown in Figure 3a, there are two types of primary carbides, which can be identified by the color. Furthermore, the compositions of 20 primary carbides of each type were detected by EDS as illustrated in Figure 3b. The results indicate that the primary carbide with white color is rich in molybdenum (Mo-PC), while the primary carbide with gray color is rich in vanadium (V-PC). Additionally, the unevenly distributed precipitations around the primary carbides are the secondary carbides.

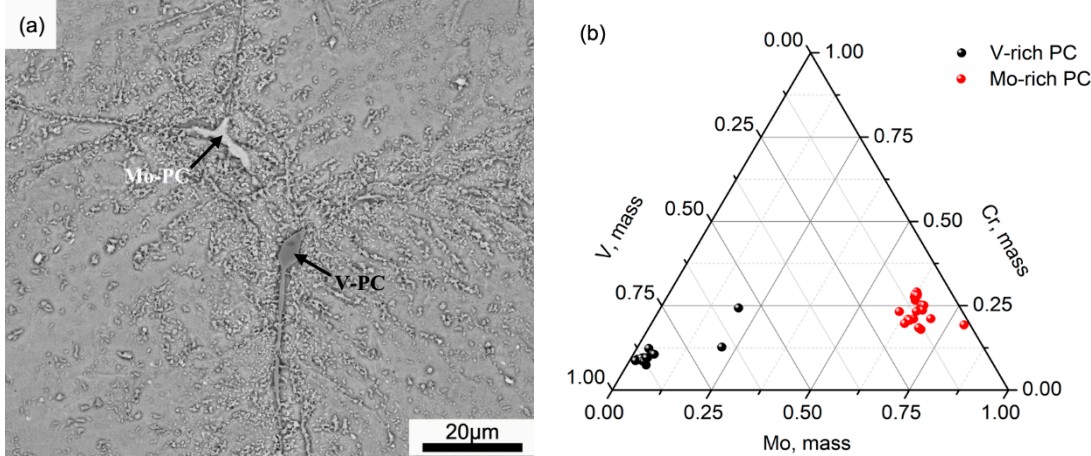

**Figure 3.** Typical primary carbides in H13 ESR ingot: (**a**) morphology of V-rich primary carbide (V-PC) and Mo-rich primary carbide (Mo-PC), and (**b**) composition distribution of V-rich PC and Mo-rich PC (online version in color).

### 3.2. Distribution of Primary Carbide

Figure 4a shows the SEM image of the H13 steel matrix. It is clear to see a bright net-like structure and the block-shaped or strip carbides (marked by arrows). Figure 4b,c demonstrates a magnifying observation of the net-like structure and strip carbide. The bright net-like structure consists of a large number of globular secondary carbide particles. The strip carbide surrounded by those secondary carbides is the primary carbide. During the solidification of the H13 steel, the solute elements, such as C, S, V, and Mo, would be enriched in the liquid phase within the interdendritic region. As the results of the enrichment of C and the strong carbide-formation elements, the primary carbide could directly precipitate from the residual liquid phase within the interdendritic region. A great deal of secondary carbides would then precipitate from the solidified steel within the interdendritic region, creating the bright net-like structure.

From the above discussion, it can be seen that the primary carbides mainly distribute in the interdendritic region, which indicates that the distribution of the primary carbide would be determined by the solidification structure of the H13 ESR ingot.

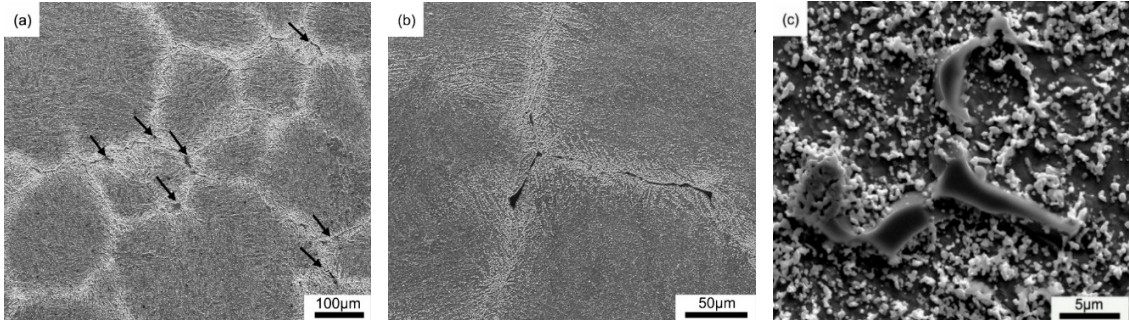

**Figure 4.** Distribution of primary and secondary carbides: (**a**) net-like structure in the matrix of sample S1, (**b**) magnifying observation of the net-like structure, and (**c**) primary carbide surrounded by secondary carbides (online version in color).

### 3.3. Microstructure of H13 ESR Ingot

Figure 5 illustrates the microstructures of the six samples of the H13 ESR ingot. The dark area implies the dendritic structure which solidifies prior to the interdendritic liquid steel as illustrated by the bright area. The cooling rate of different parts can be evaluated by following equation [17]:

$$\lambda_s = 143.9 \times R_c^{-0.3616} \times [\%C]^{(0.5501 - 1.996[\%C])}, \ [\%C] \geq 0.15 \tag{1}$$

where $\lambda_s$ is the secondary dendrite arm spacing. The secondary dendrite arm spacing of samples S1 to S6 are measured as 186, 142, 116, 115, 103, and 95 µm, respectively. The cooling rates are calculated as 0.95, 2.0, 3.5, 3.6, 4.9, and 6.1 °C/s

At the beginning of the remelting, the heat of the molten steel pool would be transferred to both the bottom and the lateral wall of the water-cooled mold. With the growing of the solidified ingot, the heat resistance along the vertical direction becomes larger. The cooling effect of the lateral wall thus outweighs that of the bottom. The heat would be mainly taken away along the radial direction. As a result, the cooling efficiency dramatically decreases with the ESR process. The dendritic grains at the upper part of the ESR ingot (S1, S2, and S3) become coarser than that at the lower part of the ESR ingot (S4, S5, and S6). Moreover, the sizes of the dendritic grain at the center of the ingot (S1 and S4) are also larger than that at the outer edge of the ingot (S3 and S6) because of the later solidification, especially at the upper part of the ingot. It is obvious that the difference of the dendritic grain size between samples S1 and S4 is greater than that between samples S3 and S6.

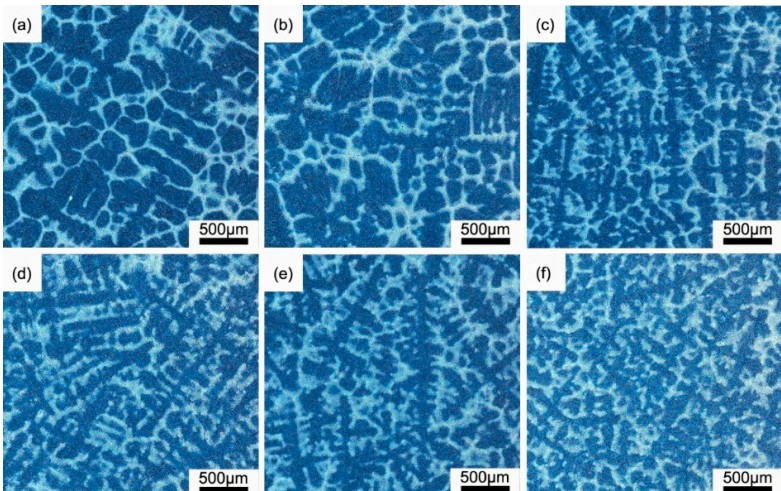

**Figure 5.** Microstructure of the H13 ESR ingot observed by optical microscopy: (**a**) sample S1, (**b**) sample S2, (**c**) sample S3, (**d**) sample S4, (**e**) sample S5, and (**f**) sample S6 (online version in color).

### 3.4. Size and Area of Primary Carbide

Fifty of randomly-selected SEM visual fields at 2000 magnification with a total area of 0.845 mm$^2$ for each sample were employed to count the size, number and area of the two type primary carbides by commercial software Image-Pro Plus. Figure 6 represents the distribution of the size, number, and area fraction (ratio of the total area of the primary carbides to total area of the 50 visual fields) of the V-rich and Mo-rich primary carbides of each sample. It can be found that the quantity of the primary carbides would decrease from the ingot center to the ingot outer surface. The total number of the V-rich primary carbide decreases from 62 to 49 if we compare the upper center (S1) with the upper outer edge (S3) of the ingot, and also decreases from 59 to 48 when we compare the lower center (S4) with the lower outer edge (S6) of the ingot. The change rule of the quantity of the Mo-rich primary carbide is the same as pointed out in Figure 6c.

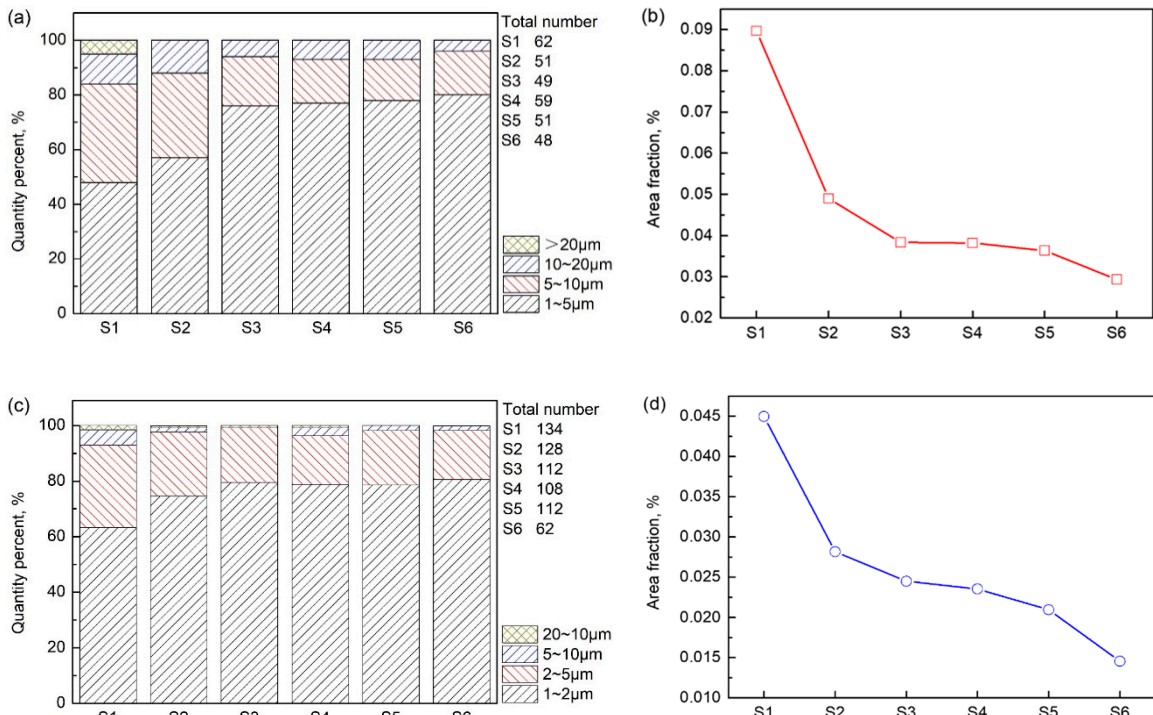

**Figure 6.** Statistical results of the primary carbides of each sample: (**a**,**b**) distribution of size and area fraction of V-rich primary carbide; (**c**,**d**) distribution of size and area fraction of Mo-rich primary carbide.

The quantity percent of the V-rich primary carbide with the size larger than 5 μm in sample S1 is about 51.6%, while the quantity percent of the Mo-rich primary carbide larger than 5 μm is only around 7.0%. In addition, the V-rich primary carbide larger than 20 μm can be only found in sample S1 (the upper center of the ingot). With the position of the sample varies from the center to the outer surface of the ingot (S1 to S3 and S4 to S6), the proportion of the primary carbides larger than 5 μm decreases. When we compare the upper part of the ingot with the lower part, it is obvious that the number of the large primary carbides also reduces, especially the samples S1 and S4. The same variation tendency can be found in the area fraction of both the V-rich and the Mo-rich primary carbides. The results indicate that the size and area fraction of the primary carbide present a decreasing trend with the increasing of the cooling rate as calculated above.

### 3.5. Heterogeneous Nucleation of Primary Carbide

The primary carbide can not only precipitate alone, but also precipitate with the non-metallic inclusions. Two kinds of non-metallic inclusion, MnS and Al$_2$O$_3$ inclusions, are commonly observed in the H13 ESR ingot. Additionally, CaO·Al$_2$O$_3$ inclusion could be occasionally detected. It can be

seen from Figure 7a that the complex precipitation consists of the V-rich primary carbide, Mo-rich primary carbide and MnS inclusion, and the complex precipitation in Figure 7b consists of the V-rich primary carbide, Mo-rich primary carbide, MnS and Al$_2$O$_3$ inclusions. The MnS inclusion is frequently observed to be the core of the Mo-rich primary carbide (Figure 7a,b). The Al$_2$O$_3$ inclusion however is more often noticed to be the core of the V-rich primary carbide (Figure 7b).

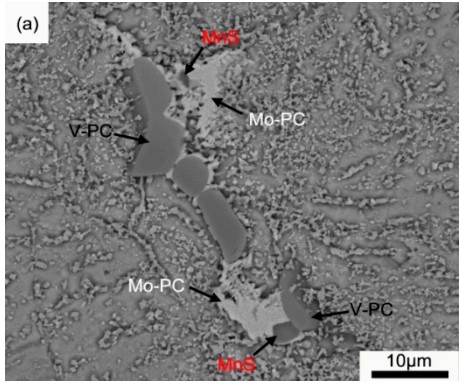 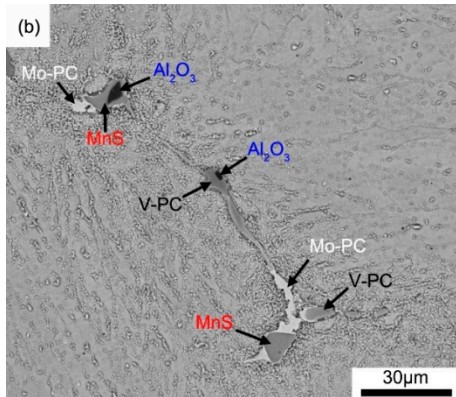

**Figure 7.** Morphology of the complex precipitations: (**a**) MnS inclusion-contained primary carbide, and (**b**) MnS and Al$_2$O$_3$ inclusion-contained primary carbide (online version in color).

The disregistry between the nucleated solid and the nucleating agent is used to describe the ability whether the nucleating agent (substrate) is effective in promoting the nucleation of the nucleated solid [10,16]. The Turnbull-Vonnegut equation is usually employed to calculate the disregistries between those phases that have similar atomic arrangements [10]. Nevertheless, the accuracy of the calculated results is out of acceptable range when it comes to the phases that have different atomic arrangements. Bramfitt modified the Turnbull-Vonnegut equation and made it work well with different atomic arrangements situation. The modified equation is shown as [18]:

$$\delta_{(hkl)_n}^{(hkl)_s} = \sum_{i=1}^{3} \frac{\left| d[uvw]_s^i \cdot \cos\theta - d[uvw]_n^i \right|}{3 \times d[uvw]_n^i} \times 100\% \tag{2}$$

where the subscripts *s* and *n* stand for the substrate and nucleated solid, respectively. (*hkl*) represents the low-index plane of the nucleating agent or nucleated solid, and [*uvw*] denotes the low-index direction in (*hkl*). *d*[*uvw*] means the interatomic spacing along the [*uvw*]. θ is the angel between [*uvw*]$_s$ and [*uvw*]$_n$. The nucleating agent is potent when the disregistry is below 12% and its potency becomes poor as it is more than 12%.

In order to explain why the V-rich primary carbide is in favor of the nucleation on the Al$_2$O$_3$ inclusion and the Mo-rich primary carbide tends to precipitate on the MnS inclusion, the disregistries between the primary carbide and the corresponding inclusion were calculated. Here, the V-rich primary carbide is identified as VC and Mo-rich primary carbide is identified as Mo$_2$C [14]. The parameters are shown in Table 2, and the matching model refers to [19]. Due to the large lattice constant differences along a certain direction, the lattice constant of the substrate or the nucleated solid should be modified in the calculation [10]. Here the lattice constant was adjusted according to the calculation condition which is shown together with the calculated results in Table 3.

**Table 2.** Lattice constants of the studied phases.

| Phase | VC | Mo$_2$C | Al$_2$O$_3$ | MnS |
|---|---|---|---|---|
| Lattice type | NaCl structure | Hexagonal crystal system | Hexagonal crystal system | NaCl structure |
| Lattice constant, nm | a1 = 0.4182 | a2 = 0.3 | a3 = 0.4812 | a4 = 0.522 |

**Table 3.** Disregistry between the studied phases.

| $\delta_n^s$ | Match Plane | Modified Lattice Constant | Disregistry, % |
|---|---|---|---|
| $\delta_{VC}^{Al_2O_3}$ | $Al_2O_3$ (0001) // VC (110) | $2 \times a1/[001]_{VC}$ | 9.4% |
| $\delta_{Mo_2C}^{Al_2O_3}$ | $Al_2O_3$ (0001) // $Mo_2C$ (0001) | $2 \times a2/[\bar{1}010]_{Mo_2C}$ | 19.8% |
| $\delta_{VC}^{MnS}$ | MnS (110) // VC (100) | $2 \times \frac{\sqrt{2}}{2}a1/[\bar{1}11]_{VC}$ | 14.6% |
| $\delta_{Mo_2C}^{MnS}$ | MnS (110) // $Mo_2C$ (0001) | $2 \times a2/[\bar{2}110]_{Mo_2C}$ | 7.9% |
| $\delta_{Mo_2C}^{VC}$ | VC (110) // $Mo_2C$ (0001) | $2 \times a2/[\bar{2}110]_{Mo_2C}$ | 8.6% |

The disregistry between the $Al_2O_3$ inclusion and the VC precipitation is 9.4% and between the MnS inclusion and the VC precipitation is 14.6%, which indicates that the $Al_2O_3$ inclusion more effective in promoting the nucleation of the VC precipitation than the MnS inclusion. The disregistry between the $Al_2O_3$ inclusion and the $Mo_2C$ precipitation is 19.8%. Such a large disregistry implies that the $Al_2O_3$ inclusion has a poor potency in promoting the nucleation of the $Mo_2C$ precipitation. Meanwhile, the disregistry between the MnS inclusion and the $Mo_2C$ precipitation is 7.9%, which reveals that the MnS inclusion is a better nucleation agent for the $Mo_2C$ precipitation. Additionally, the VC precipitation also has a low disregistry of 8.6% with the $Mo_2C$ precipitation.

Although it is in a small probability, the MnS inclusion could act as the core of the V-rich primary carbide (Figure 8a). The $Al_2O_3$ inclusion would become the core of the $Mo_2C$ precipitation only when it is in the complex inclusion which has the $Al_2O_3$ core and MnS outer layer (Figure 8c). Nevertheless, no CaO·$Al_2O_3$ inclusion, as shown in Figure 8e, is found to serve as the nucleation site of the primary carbide.

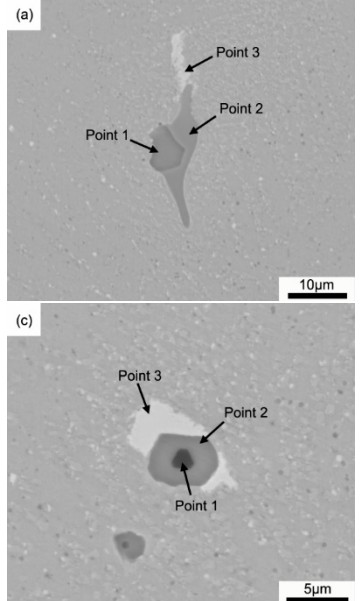 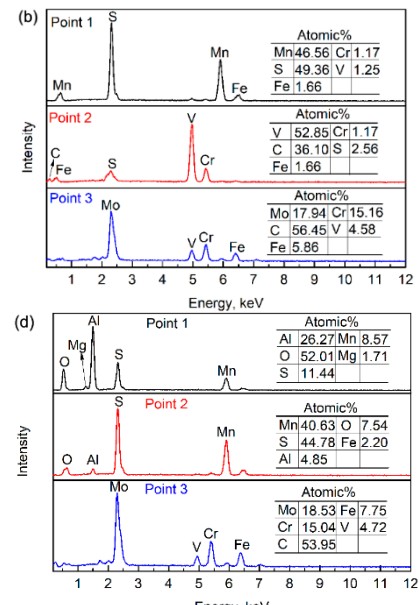

**Figure 8.** *Cont*.

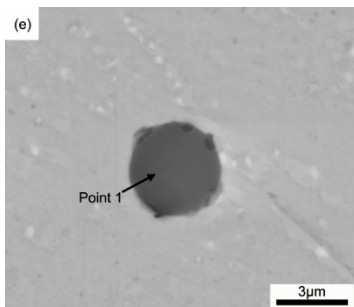
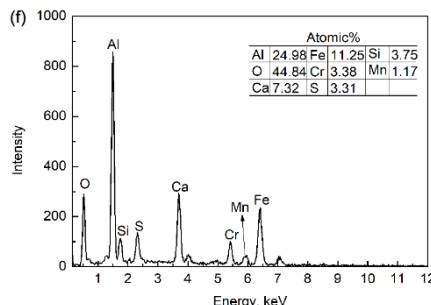

**Figure 8.** SEM images and compositions of primary carbides and inclusions: (**a,b**) MnS inclusion containing primary carbide, (**c,d**) MnS and $Al_2O_3$ inclusions containomg primary carbide, and (**e,f**) $CaO \cdot Al_2O_3$ inclusion (online version in color).

Except for the disregistry, the substrate should be in solid state when it acts as the nucleation site. The formation reaction of the $Al_2O_3$ inclusion, MnS inclusion, VC and $Mo_2C$, as well as the equilibrium constant $K$, are described as follows [14,20,21]:

$$2[Al] + 3[O] = Al_2O_3(s) \quad \log K = 64000/T - 20.57 \tag{3}$$

$$[Mn] + [S] = MnS(s) \quad \log K = 8630/T - 4.75 \tag{4}$$

$$[V] + [C] = VC(s) \quad \log K = 5438/T - 4.09 \tag{5}$$

$$2[Mo] + [C] = Mo_2C(s) \quad \log K = 6454/T - 7.47 \tag{6}$$

It can be seen from Equations (3)–(6) that the $Al_2O_3$ inclusion would be formed before the taking place of the solidification of the investigated H13 steel. However, the concentrations of C, S, Mn, V, and Mo of the H13 ingot (Table 1) are far from satisfying the formation condition of the MnS inclusion and the primary carbide.

Due to the segregation during the solidification, the solute concentration would vary with the solid fraction which makes it possible for the generation of the MnS inclusion and the primary carbide. The concentration of the solute can be evaluated by the Clyne-Kurz equation, defined as:

$$C_L = C_0[1 - (1 - 2\Omega(\alpha)k)f_S]^{\frac{k-1}{(1-2\Omega(\alpha))k}} \tag{7}$$

$$\Omega(\alpha) = \alpha\left[1 - \exp\left(-\frac{1}{\alpha}\right)\right] - \frac{1}{2}\exp\left(-\frac{1}{\alpha}\right) \tag{8}$$

$$\alpha = \frac{D_S t_f}{(0.5\lambda)^2} \tag{9}$$

$$t_f = \frac{T_{\updownarrow} - T_S}{R_C} \tag{10}$$

where $C_L$ is local concentration of the element in the liquid phase, and $C_0$ is the initial concentration. $k$ and $D_S$ are the equilibrium partition coefficient and diffusion coefficient of the element between the liquid phase and solid phase, respectively (Table 4) [14,22]. $f_S$ is the solid fraction and $t_f$ is the local solidification time. $T_{\updownarrow}$ and $T_s$ are the liquidus and solidus temperatures of the H13 steel, respectively. The values of $T_{\updownarrow}$ and $T_s$ can be determined [23,24]:

$$
\begin{aligned}
T_{\updownarrow}(K) = \ & 1809 - \big\{100.3[\%C] - 22.4[\%C]^2 - 0.16 + 13.55[\%Si] \\
& -0.64[\%Si]^2 + 5.82[\%Mn] + 0.3[\%Mn]^2 \\
& +4.2[\%Cu] + 4.18[\%Ni] + 0.01[\%Ni]^2 \\
& +1.59[\%Cr] - 0.007[\%Cr]^2\big\}
\end{aligned}
\tag{11}
$$

$$T_s(K) = \begin{aligned} &1809 - \{415.5[\%C] + 12.3[\%Si] + 6.8[\%Mn] \\ &+124.5[\%P] + 183.9[\%S] + 4.3[\%Ni] \\ &+1.4[\%Cr] + 4.1[\%Al]\} \end{aligned} \qquad (12)$$

**Table 4.** Equilibrium partition coefficients ($k_i$) and diffusion coefficients ($D_S$) of the studied elements.

| Element | $k_i$ | $D_S$, cm$^2$/s |
|---------|-------|-----------------|
| Mn | 0.785 | $0.055\exp(-249378/RT)$ |
| S | 0.035 | $2.4\exp(-223426/RT)$ |
| C | 0.34 | $0.0768\exp(-143511/RT)$ |
| V | 0.63 | $0.284\exp(-258990/RT)$ |
| Mo | 0.585 | $0.068\exp(-246856/RT)$ |

The calculated liquidus and solidus temperatures of the H13 steel in the present work are 1749 K and 1613 K, respectively.

Taking the formation reaction of the MnS inclusion as an example, the equilibrium concentration product of [Mn] and [S] can be expressed as:

$$\log K = \log\frac{a(\text{MnS})}{a[\text{Mn}] \cdot a[\text{S}]} = \log\frac{a(\text{MnS})}{[\text{Mn}] \cdot [\text{S}]} - \log f_{\text{Mn}} - \log f_{\text{S}} \qquad (13)$$

where the activity of the MnS inclusion is unity (regarding pure solid MnS as the standard state). $f_{\text{Mn}}$ and $f_{\text{S}}$ are the activity coefficients of dissolved elements Mn and S, respectively. The activity coefficients $f_i$ with different temperatures can be calculated [25]:

$$\log f_i = e_i^i[\%i] + e_i^j[\%j] + \cdots \qquad (14)$$

$$e_{i(T)}^j = \left(\frac{2538}{T} - 0.355\right) \times e_{i(1873K)}^j \qquad (15)$$

where $e_i^i$ and $e_i^j$ are the interaction coefficients of activity. The values of the interaction coefficients of activity at 1873 K are collected from [25]. $T$ is the temperature of solidification front and can be expressed as [26]:

$$T = T_0 - \frac{\left(T_0 - T_\updownarrow\right)}{1 - f_S\frac{(T_\updownarrow - T_s)}{(T_0 - T_s)}} \qquad (16)$$

where $T_0$ is the melting point of pure iron (1809 K).

Figure 9 indicates the variation of the element concentration product with the solid fraction. The MnS inclusion or the primary carbide would not precipitate unless the actual concentration product is larger than the equilibrium concentration product. It can be seen that the V-rich primary carbide would precipitate when the solid fraction ranges from 0.965 to 0.983 in the studied cooling rates. The MnS inclusion would be formed if the solid fraction is 0.989–0.990. Note that although Mo dominates the Mo-rich primary carbide, there are still large parts of other alloy elements like Cr and V, the activity of Mo$_2$C may not be equal to 1 [13]. The atomic ratio of Mo to the alloy elements in the carbide is about 0.4. When taking the activity of Mo$_2$C as 1, the Mo$_2$C cannot precipitate. The activity, however, is assumed to be 0.4, the Mo-rich primary carbide would, thus, precipitate when the solid fraction ranges from 0.9990 to 0.9998.

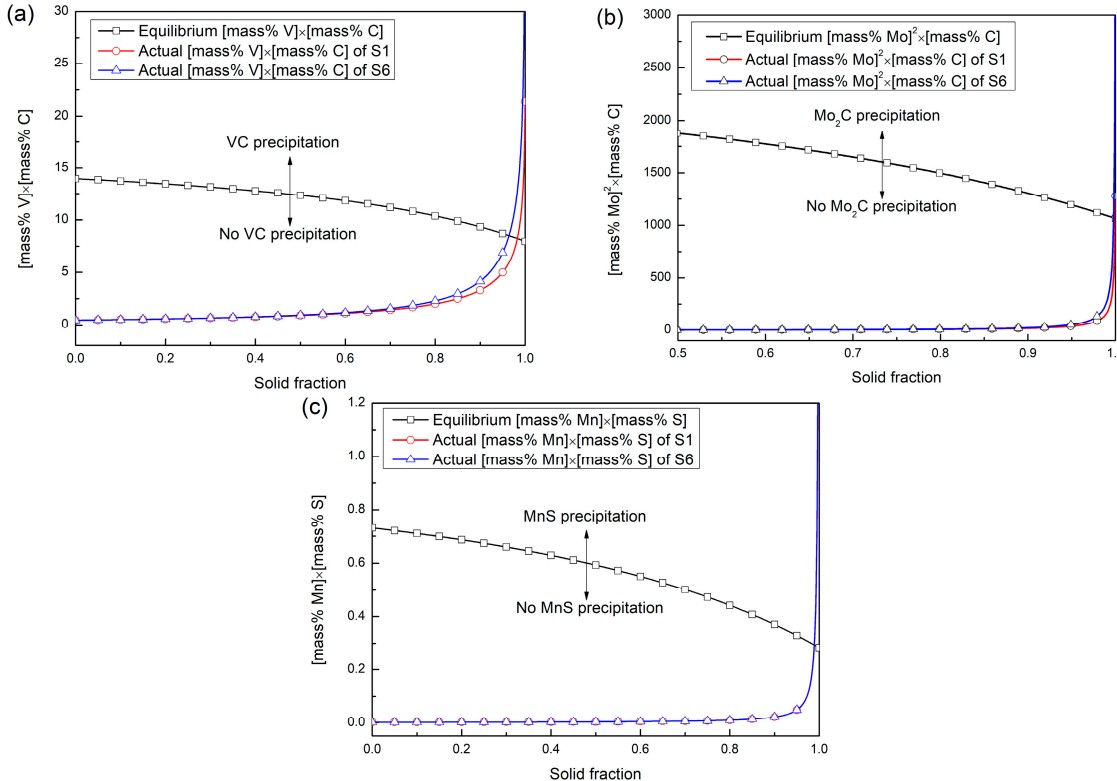

**Figure 9.** Relationship between element concentration product and solid fraction: (**a**) elements V and C, (**b**) elements Mo and C, and (**c**) elements Mn and S (online version in color).

It can be inferred that the $Al_2O_3$ inclusion would be formed in the molten steel before the solidification occurs. Following is the precipitation of the V-rich primary carbide, which is easier to precipitate on the $Al_2O_3$ inclusion for their low disregistry. Then the MnS inclusion would be generated during the solidification, the Mo-rich primary carbide would precipitate at last. The MnS inclusion as well as the V-rich primary carbide could act as the effective nucleating agent for the Mo-rich primary carbide as discussed above, but the $Al_2O_3$ inclusion cannot play the role of the nucleation site unless it exists in the form of $Al_2O_3$-MnS complex inclusion. The $CaO \cdot Al_2O_3$ inclusion would be prevented from being the nucleation site for the primary carbide precipitation, since this oxide inclusion has a low melting point and unmatched interfacial energy [16].

## 4. Conclusions

In order to improve the performance of H13 steel, it is essential to understand the precipitation of the primary carbide. In the present work, the composition, the distribution, and the size of the V-rich and Mo-rich primary carbides in a commercial-sized H13 ESR ingot were studied. The relationship between the primary carbides and the inclusions was discussed in detail. The main conclusions can be drawn as follows:

1. Two types of primary carbide, V-rich and Mo-rich primary carbides, are observed in the H13 ESR ingot, which mainly distribute in the interdendritic region.
2. A net-like structure, caused by the enrichment of the solutes within the interdendritic region, is found in the matrix of the H13 ESR ingot. The primary carbide would be formed in the center of the net-like structure. Furthermore, the dendritic grain of the lower part of the H13 ESR ingot is finer than that in the upper part, and also dendritic grain at the outer surface of the ingot is finer than that at the center of the ingot.
3. The quantity: the area fraction and the size of the two primary carbides tend to decrease from the center of the H13 ESR ingot to the outer surface. Additionally, the V-rich primary carbide is

obviously larger than the Mo-rich primary carbide. A stronger cooling would reduce the size of the primary carbides.

4.  The $Al_2O_3$ inclusion can promote the nucleation of the V-rich primary carbide, while the MnS inclusion can encourage the nucleation of the Mo-rich primary carbide. The solid fraction that the V-rich begins to precipitate in the investigated steel ranges from 0.965 to 0.983, and that for the Mo-rich primary carbide and the MnS inclusion change from 0.9990 to 0.9998 and from 0.989 to 0.990, respectively. The $CaO·Al_2O_3$ inclusion cannot act as the nucleation site for the precipitation of the two primary carbides.

**Author Contributions:** G.L., Q.W. and F.W. conceived and designed the experiments; X.W., Y.L. and Y.C. performed the experiments; X.W., Y.L., Y.C. and F.W. analyzed the data; X.W. wrote the first draft of the manuscript; Q.W. revised and approved the final version of the manuscript. Investigation, Y.L. and Y.C.; Project administration, Q.W.

**Funding:** This research was funded by the National Natural Science Foundation of China (grant no. 51804227).

**Acknowledgments:** The authors' gratitude goes to the National Natural Science Foundation of China (grant no. 51804227). The industrial experiment was also supported by Hubei Rising Technology Co. Ltd., in Huangshi City, Hubei Province, China.

**Conflicts of Interest:** The authors declare no conflict of interest.

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
