# Peer review of "Investigation of Primary Carbides in a Commercial-Sized Electroslag Remelting Ingot of H13 Steel"

_metals, doi:10.3390/met9121247_

Round 1
Reviewer 1 Report
In my opinion the paper is well-prepared and brings one whole, logic story, bringin novelty to the processing of H13 steel. In my opinion, size of your "sample" is impressive!
I'd like to ask is it possible, in Figs 4, 5, 7 to supplement by relevant EDS elemental maps? It would be nicer for reader, however points analyses presented by the Authors are already bringing valuable information.
In my opnion, the paper deserved to be published in MDPI's Metals.
Author Response
Q: I'd like to ask is it possible, in Figs 4, 5, 7 to supplement by relevant EDS elemental maps? It would be nicer for reader, however points analyses presented by the Authors are already bringing valuable information.
A: Thank you for your suggestion. We are sorry that we do not have the elemental maps because we only analyzed the composition of these points by EDS.

Reviewer 2 Report
…primary carbide The…
Should be:
…primary carbide. The…
In order to remove the moisture, the slag powder was heated for 4 hours at 1123 K in a muffle furnace.
This phrase needs to be improved, because the moisture can be removed at much lower temperatures.
Table 1
Please define “T. O”
Figure 3 (b).
The mass compositions of V, Cr and Mo are varied from 0 % to 1 %. Please check if these values have to be between 0 % and 100 %. Modify, if necessary.
According to Table 2 (Lattice constants of the studied phases), the lattice type of VC and MnS are identical (NaCl structure). Likewise, the lattice type of Mo2C and Al2O3 are also identical (hexagonal crystal system). Hence, one may envisage that MnS inclusions can serve as nucleation for the VC phase. Likewise, the Al2O3 can help the nucleation and/or growth of Mo2C phase.
Please check also this hypothesis, and make comments, if necessary.
Furthermore, write “NaCl” instead of “Nacl” (see Table 2).
The values of carbon contents (point 3 with a Carbon content of about 55 %) as shown in Figure 8 seem to be so high compared with the carbon peak of the SEM-EDS spectra. Please check and modify, if necessary. Is really SEM-EDS a good tool for carbon quantification?
Please add "doi" of the article (in the Reference list), where possible.
Author Response
Q1:“…primary carbide The…” Should be: …primary carbide. The… .
A1:Thank you for correcting our mistake. This part has been revised in our modified manuscript.
Q2:In order to remove the moisture, the slag powder was heated for 4 hours at 1123 K in a muffle furnace.This phrase needs to be improved, because the moisture can be removed at much lower temperatures.
A2:Thank you for your suggestion. We have changed the sentence into “The slag powder was heated for 4 hours at 1123 K in a muffle furnace to ensure the removal of moisture from slag.” in our modified manuscript.
Q3:Table 1, Please define “T. O”.
A3:Thank you for your suggestion. T. O is the total oxygen in steel. We have added the definition of T. O in the modified manuscript.
Q4: Figure 3 (b).
The mass compositions of V, Cr and Mo are varied from 0 % to 1 %. Please check if these values have to be between 0 % and 100 %. Modify, if necessary.
A4: Thank you for correcting our mistake. The compositions of V, Cr and Mo in Figure 3 (b) were supposed to be the mass ratio, however were mismarked as mass percent. This part has been modified in our revised manuscript.
Q5:According to Table 2 (Lattice constants of the studied phases), the lattice type of VC and MnS are identical (NaCl structure). Likewise, the lattice type of Mo2C and Al2O3 are also identical (hexagonal crystal system). Hence, one may envisage that MnS inclusions can serve as nucleation for the VC phase. Likewise, the Al2O3 can help the nucleation and/or growth of Mo2C phase.
Please check also this hypothesis, and make comments, if necessary.
Furthermore, write “NaCl” instead of “Nacl” (see Table 2).
A5:Thank you for your suggestion. Mo2C and Al2O3 do have the same lattice type. However, the disregistry between Mo2C and Al2O3 (19.8%) is higher than 12% even after modifying the lattice constant according to our calculation (Table 3), which means Al2O3 may be not effective enough to promote the nucleation of Mo2C. Besides, the disregistry between VC and Al2O3 (9.4%) is lower than 12% and that implies Al2O3 is effective to promote the nucleation of VC. According to Figure 9, Mo2C precipitates much later than VC. Thus, Al2O3 tends to be the core of VC, which also prevent Al2O3 from helping the nucleation of Mo2C. In fact, the single phase Al2O3 was scarcely observed to be the core of Mo2C in our experiment.
Although the disregistry between VC and MnS (14.6%) is larger than that between Mo2C and MnS (7.9%), VC is formed much earlier than Mo2C as mentioned above, and that makes it possible for MnS to be the core of VC. Furthermore, we did find a few complex precipitations consisting of MnS and VC like Figure 8 (a).
Finally, thank you for correcting our mistake. We have changed the “Nacl” in Table 2 into “NaCl” in our modified manuscript.
Q6:The values of carbon contents (point 3 with a Carbon content of about 55 %) as shown in Figure 8 seem to be so high compared with the carbon peak of the SEM-EDS spectra. Please check and modify, if necessary. Is really SEM-EDS a good tool for carbon quantification?
A6:Thank you for your suggestion. The atomic percent of carbon in Mo-rich primary carbide is about 55 %, and weight percent of that is about 16% which is very close to that in literature: Mao, M.T.; Guo, H.J.; Wang, F.; Sun, X.L. Effect of cooling rate on the solidification microstructure and characteristics of primary carbides in H13 steel. 2019, 59, 848-857. The reason why the carbon peak of the SEM-EDS spectra is not obvious may be because the carbon is light element compared with other alloying elements. Besides, the range of the Y-axis (Intensity) was set relatively large for including three curves in one picture.
Q7:Please add "doi" of the article (in the Reference list), where possible.
A7:Thank you for your suggestion. The DOI of these articles is attached to the end of corresponding articles.
